# Synthesis, Antimicrobial Activity, and Tyrosinase Inhibition by Multifunctional 3,4-Dihydroxy-Phenyl Peptidomimetics

**DOI:** 10.3390/ijms26041702

**Published:** 2025-02-17

**Authors:** Deepak S. Wavhal, Dominik Koszelewski, Paweł Kowalczyk, Anna Brodzka, Ryszard Ostaszewski

**Affiliations:** 1Institute of Organic Chemistry, Polish Academy of Sciences, Kasprzaka 44/52, 01-224 Warsaw, Poland; deepak.wavhal@icho.edu.pl (D.S.W.); dominik.koszelewski@icho.edu.pl (D.K.); anna.brodzka@icho.edu.pl (A.B.); 2Department of Animal Nutrition, The Kielanowski Institute of Animal Physiology and Nutrition, Polish Academy of Sciences, Instytucka 3, 05-110 Jabłonna, Poland; p.kowalczyk@ifzz.pl

**Keywords:** tyrosinase, minimum inhibitory concentration (MIC), gram-positive bacteria, gram-negative bacteria, multicomponent reaction, Ugi reaction

## Abstract

The purpose of the present study was to evaluate the synergistic effect of two important pharmacophores, 3,4-dihydroxyphenyl and peptidomimetic moieties, as mushroom tyrosinase inhibitors and antimicrobial agents targeting specific strains of pathogenic bacteria. The 3,4-dihydroxybenzaldehyde (protocatechuic aldehyde) was found to be an effective inhibitor of tyrosinase activity, and due to the fact that it is a safe natural substance with such a scaffolded structure, it is likely that dihydroxyl-substituted phenolic derivatives can exhibit potent tyrosinase inhibitory activity. Series of peptidomimetics with an incorporated 3,4-dihydroxyphenyl scaffold was synthesized and characterized. The inhibitory effects of peptidomimetics on a mushroom tyrosinase were studied. The results showed that among the compounds, five of them showed higher inhibitory activity than the parent 3,4-dihydroxybenzyl aldehyde. In silico docking studies with mushroom tyrosinase (PDB ID 2Y9X) predicted possible binding modes in the enzymatic pocket for these compounds. Furthermore, the antimicrobial activities of peptidomimetics against selected Gram-positive and Gram-negative bacterial strains (*E. coli*, *A. baumannii*, *P. aeruginosa*, *E. cloacae*, and *S. aureus*) were investigated. The results showed that all tested peptidomimetics have antimicrobial activities (MIC values from 0.25 to 4.0 μM) comparable with those observed for the commonly used antibiotics (ciprofloxacin, bleomycin, and cloxacillin). Notably, all evaluated compounds demonstrated significant activity against *E. coli* and *S. aureus* strains, which are primary sources of infections resulting in numerous fatalities. Additionally, the cytotoxicity of sixteen derivatives was assessed using the MTT assay on BALB/c3T3 mouse fibroblast cell lines. Cytotoxicity analyses indicated that the tested substances have a similar or reduced impact on cell proliferation compared to commonly utilized antibiotics within the range of therapeutic doses. This study presents the potential of peptidomimetics with 3,4-dihydroxyphenyl scaffolds could be beneficial for developing novel tyrosinase inhibitors and new potent food preservatives or cosmetic additives.

## 1. Introduction

According to a recent study in 2024, common bacterial infections were associated with 4.71 million deaths, including 1.14 million directly attributable deaths, with the burden shifting significantly toward older adults over the past three decades. While AMR mortality decreased by over 50% in children under five, it increased by more than 80% in adults aged 70, underscoring an urgent need for targeted interventions [1]. According to the World Health Organization (WHO) report, over 200 diseases are caused by eating food contaminated with bacteria and other pathogens, causing significant socioeconomic impact through strains on healthcare systems, lost productivity, and harming tourism and trade. These infections contribute significantly to the worldwide burden of disease and mortality. Foodborne diseases are caused by food contamination and arise at any stage of the food manufacture, distribution, and consumption chain. The preservation of food is crucial to improve the shelf life of food and guarantee food safety. Enzymatic browning and microbial infection seriously affect the quality of fresh-cut fruits and vegetables. For these reasons, food preservation is based mainly on controlling enzymes or microbial deteriorative processes. Anti-tyrosinase and antimicrobial activities are critical areas of research with wide-reaching implications in industries such as food preservation, cosmetics, and pharmaceuticals. Tyrosinase, a copper-containing enzyme, plays a crucial role in the biosynthesis of melanin, a process essential for pigmentation in humans and responsible for browning fruits and vegetables [2]. On human skin, the overactivity of tyrosinase can cause hyperpigmentation disorders, including melasma, age spots, and post-inflammatory hyperpigmentation. These conditions often require cosmetic interventions that involve the inhibition of tyrosinase to prevent excessive melanin production [3]. In food products, tyrosinase catalyzes the oxidation of phenolic compounds, leading to enzymatic browning, resulting in undesirable changes in appearance, texture, and taste. Therefore, tyrosinase inhibitors are valuable both in the cosmetic industry for depigmentation treatments and in the food industry to prevent browning during storage and processing [4]. On the other hand, it refers to the ability of a compound to inhibit or kill microorganisms, including bacteria, fungi, and viruses. The rise in antimicrobial resistance (AMR) has led to an urgent search for novel antimicrobial agents that can effectively combat resistant pathogens [5]. Traditionally, antibiotics have been aimed at specific bacterial processes, but the overuse of these drugs has led to the emergence of multidrug-resistant strains. Natural antimicrobial peptides (AMPs) are part of the innate immune system and offer broad-spectrum antimicrobial activity. However, its clinical potential is limited by rapid degradation in biological systems and short half-lives. This has led to the development of peptidomimetics, synthetic compounds designed to mimic the structure and function of AMPs while offering improved stability, longer half-lives, and resistance to proteolytic enzymes [6]. Peptidomimetics are small peptide-based molecules designed to replicate the physicochemical characteristics (such as structure and hydrophobicity) and biological functions (such as antimicrobial activity and mechanism of action) of antimicrobial proteins (AMPs). These compounds often exhibit enhanced resistance to proteolytic degradation and extended in vivo half-lives due to the incorporation of an unnaturally structured backbone [7]. Recent studies have explored the antibacterial properties of tyrosinase inhibitors, some demonstrating superior efficacy compared to existing antibiotics [8]. Bacterial melanin plays a role in enhancing long-term cell viability by protecting bacteria from UV radiation and chelating metals, thus helping bacterial survival under stress [9]. Furthermore, melanin can neutralize antibiotics, reducing their effectiveness and allowing bacteria to thrive in the presence of drugs such as polymyxin B, kanamycin, tetracycline, and ampicillin. Consequently, the inhibition of tyrosinase can reduce melanin production, potentially restoring or improving antibiotic efficacy [10,11,12]. Given that melanin also increases bacterial virulence, these findings highlight the potential for tyrosinase inhibitors to contribute to the development of new antimicrobial agents and improve the effectiveness of existing antibiotics, especially in the face of increasing antibiotic resistance [13]. Moreover, microbial dysbiosis is linked with chronic inflammation and inflammation-mediated carcinogenesis processes. Chronic exposure to inflammatory mediators produced by bacteria can lead to increased cell proliferation and cancer [14].

In recent decades, numerous potential tyrosinase inhibitors have been identified; however, many of these compounds are not suitable for therapeutic use due to their pharmacological properties. A common characteristic among many of these inhibitors is the presence of hydroxy groups in their chemical structures. For example, well-known tyrosinase inhibitors available on the market, such as kojic acid and arbutin, also contain multiple hydroxy moieties. Conversely, many of these compounds struggle to penetrate cellular barriers effectively. Based on our previous research, we have observed that peptidomimetics—compounds that mimic natural peptides in the body—exhibit the potential to traverse these barriers. Consequently, we designed several peptidomimetics that incorporate dihydroxy groups. To facilitate this synthesis, we utilized the established tyrosinase inhibitor 3,4-dihydroxybenzaldehyde as a precursor [15] (Figure 1). This compound is structurally isosteric with L-DOPA, a natural substrate for tyrosinase involved in the biosynthesis of melanin. By incorporating 3,4-dihydroxybenzaldehyde, our aim is to competitively inhibit tyrosinase activity using its structural similarity to L-DOPA. We employed the Ugi reaction, a powerful synthetic tool, to synthesize these peptidomimetics containing dihydroxy groups. The goal of this study is to synthesize and evaluate peptidomimetics that exhibit dual functionality, providing both tyrosinase inhibition and antimicrobial properties. By combining these activities into a single compound, we aim to create multifunctional agents with potential applications in food preservation and the treatment of microbial skin infections. Our findings demonstrate the potential of these peptidomimetics to act as both effective antimicrobial agents and inhibitors of enzymatic browning, expanding the scope of their use in various industrial and clinical settings. Simultaneously, the peptidomimetics backbone can improve antimicrobial activity by disrupting microbial cell membranes and interfering with cellular processes [16] (Figure 1).

## 2. Results and Discussion

### 2.1. Chemistry

The target peptidomimetics were synthesized according to a recently published procedure [17]. The Ugi model reaction was conducted between 3,4-dihydroxybenzaldehyde (**1**), benzylamine (**2a**), phenylacetic acid (**3a**), and benzyl isocyanide (**4a**) in methanol at 50 °C, providing the desired peptidomimetic 5a with 68% yield (Figure 1).

Furthermore, under similar conditions, seventeen structurally diverse peptidomimetics (**5b**–**r**) were provided in moderate yields ranging from 48% to 68% (Figure 2). The structures of all the compounds obtained were confirmed using ^1^H, ^13^CNMR, and HRMS, as provided in the Appendix A. Additionally, the chemical purity of newly synthesized compounds **5a**–**r** was verified by elemental analysis.

### 2.2. Tyrosinase Inhibitory Ability

Tyrosinase (Tyr) extracted from *Agaricus bisporus* (mushroom tyrosinase, mTYR) is commonly used as a model enzyme for human Tyr due to its high purity, stability, and commercial availability. It is frequently used in the selection of TYR inhibitors [18,19,20]. Following established literature protocols, L-tyrosine and L-DOPA, which serve as substrates for monophenol and diphenol oxidation, respectively, were used to assess the tyrosinase inhibitory activity of synthesized target compounds [21]. Using L-DOPA as a substrate, compound **5e** demonstrates the highest inhibition activity of tyrosinase of 30% at a concentration of 100 µM (Figure 3).

Although this activity is lower than that of kojic acid (KA), which served as a reference and has 49% inhibition activity under the same conditions, it should be noted that compound **5e** showed greater inhibitory activity compared to the parent 3,4-dihydroxybenzaldehyde (**1**), which shows only 15% inhibition activity at 100 µM (Figure 3). The reported IC_50_ value for 3,4-dihydroxybenzaldehyde is 250 µM [15]. At a concentration of 100 µM, compounds **5a**, **5c**, **5f**, **5h**, **5j**–**n**, **5q,** and **5r** show less than 10% inhibition activity of mTyr. In contrast, compounds **5b**, **5d**, **5e**, **5g**, **5i**, **5o,** and **5p** demonstrate more than 10% inhibition activity at the same concentration.

### 2.3. Computational Study (In Silico Study)

#### 2.3.1. Molecular Docking Analysis

Molecular coupling analysis was used to investigate the binding interactions of compound **5e** with the active site of mushroom tyrosinase (mTyr). The 3D structure of mTyr, sourced from *Agaricus bisporus* (PDB ID: 2Y9X), was utilized as the docking target. The best binding conformation is presented in Figure 4, showing the fitting of compound **5e** into the cavity of the active site through multiple stabilizing interactions.

The 3,4-dihydroxy moiety of compound **5e** was oriented to occupy the spacious region around the active site, forming critical interactions with residues His259, His263, Val283, Gly281, and Asn260, in addition to coordinating with the single copper atom. In particular, the 4-methoxy group of the isocyanide moiety formed hydrogen bonds with His85 and Asn81, further stabilizing the binding. The 3,4-dihydroxy group showed a pi-sigma interaction with Val283 and also established several hydrogen bonds with His263, His259, Gly281, and copper ions, anchoring the molecule within the active site. Interestingly, Val283 participated in pi–pi T-shaped interactions with both the aromatic 3,4-dihydroxy ring and the aliphatic cyclohexyl group, improving the binding affinity through hydrophobic and π–electron interactions. These findings show that compound **5e** binds effectively within the active site, exhibiting a binding energy of −8.4 kcal/mol (Table 1), involving a combination of hydrogen bonding, pi interactions, and metal coordination. Thus, the obtained molecular docking data corresponds well with the experimental results obtained in the tyrosinase inhibition assay.

#### 2.3.2. ADMET Profiles

Pharmacokinetic parameters and toxicity data were acquired utilizing the pkCSM online tool and presented in Table 2.

The ADMET analysis conducted on various compounds, utilizing kojic acid (KA) as a benchmark, provides essential insights into their pharmacokinetic and toxicological characteristics. The compounds exhibited molecular weights conducive to effective penetrability. The Caco-2 permeability measurements ranged from 0.637 to 1.118, and intestinal absorption (IA) spanned from 70.802% to 91.477%, illustrating efficient absorption through the small intestine. Skin permeability (SP) values, which extended from −2.943 to −2.136 cm/h, indicated suitable transdermal efficacy. The volume of distribution at steady state (VDss) values was mainly negative or approximately zero, suggesting limited systemic distribution, consistent with the intent of localized therapeutic effects rather than extensive distribution. Concerning the capacity to traverse the blood–brain barrier (BBB), KA presented a log BB value of −0.086, indicating minimal central nervous system (CNS) penetration, whereas certain compounds, such as **5m**, displayed positive log BB values, suggesting potential CNS activity. The compounds generally exhibited positive total clearance values, which denote effective elimination. The acute oral toxicity (LD_50_) in rats for the compounds tested ranged from 2.037 to 2.801 mol/kg, with KA demonstrating moderate toxicity levels at 2.037 mol/kg. Evaluations of chronic toxicity revealed LOAEL values between 0.689 and 1.613 mol/kg, indicating manageable toxicity profiles for the compounds, thereby underscoring the necessity for meticulous appraisal in therapeutic contexts.

### 2.4. In Vitro Biological Studies of Synthesized Compounds

#### 2.4.1. MIC and MBC Studies

The antibacterial properties of the nineteen synthesized peptidomimetics **5a**–**r**, as shown in Figure 5 (Appendix A), were evaluated using a previously established protocol [22]. The evaluation involved determining the minimum inhibitory concentration (MIC) and the minimum bactericidal concentration (MBC) (Figure 6 and Appendix A) for various bacterial strains. The MIC values varied from 0.25 to 4.0 μM, whereas MBC values ranged from 0.5 to 4.0 (±0.5) μM. These tests were carried out on *E. coli* strains K12 (ATCC 25404), *E. coli* R2 (ATCC 39544), *E. coli* R3 (ATCC 11775), and *E. coli* R4 (ATCC 39543), as well as Gram-negative bacteria *A. baumannii* (ATCC 17978), *P. aeruginosa* (ATCC 15442), *E. cloacae* (ATCC 49141), and *S. aureus* (ATCC 23235), as illustrated in Figure 5. MIC values indicate the susceptibility or resistance of bacteria to the compounds, while MBC values determine the concentration needed to completely eliminate bacterial growth, thus distinguishing between bactericidal and bacteriostatic effects [23]. All compounds demonstrate effective antibacterial activity, with MIC values below 4 μM for both Gram-positive and Gram-negative strains (Figure 5). Additionally, the antibacterial efficacy was notably influenced by variations in the isocyanide substituents. In particular, 3,4-dihyroxybenzaldehyde (**1**) was less potent than all designed and synthesized peptidomimetic **5**. It was shown that the peptidomimetics have increased activity toward bacteria. Interestingly, compounds **5a**, **5d**–**h,** and **5n**–**r** show broad-spectrum activity toward both Gram-positive and Gram-negative bacteria. Compound **5a** shows a very potent activity against both Gram-negative and Gram-positive bacteria. Recent studies revealed that the B2 phylotype of commensal *E. coli* strains is a frequent carrier of virulence genes and often causes extraintestinal infections [24]. *E. coli*, an important extraintestinal pathogen in humans and certain animals, is a common cause of urinary tract infection, sepsis, neonatal meningitis, and colibacillosis [25,26]. Patients with inflammatory bowel disease and bowel cancers have been found with huge numbers of pathogenic *E. coli* living inside tumors. That is because these *E. coli* can stick to and invade the cells lining our bowels and replicate inside them. To make things worse, these *E. coli* are capable of producing a toxic substance called colibactin that damages the DNA of bowel cells, making them cancerous, and can help cancers to spread [27]. However, *S. aureus* and its virulence factors have been implicated in several skin diseases, and *staphylococcal* superantigens (SAgs) may contribute to the development of malignancies such as cutaneous T-cell lymphoma [28,29].

Furthermore, as illustrated in Figure 5 and Figure 7, the evaluated peptidomimetics generally demonstrated superior antibacterial activity compared to conventional antibiotics. This is particularly significant given the increasing resistance observed in microorganisms to standard antibiotics such as cloxacillin (clox) and ciprofloxacin (cipro) (Figure 7 and Appendix A). Although the development of resistance to bleomycin (bleo) remains uncertain, its emergence could significantly affect the current antibiotic regimen used to treat hospital infections. In particular, our compounds showed an antibacterial efficacy comparable to that of bleomycin (bleo) [5].

In particular, compounds that feature an acyl moiety as the acid component of the peptidomimetics exhibited reduced potency against *E. coli* strains. On the contrary, the inclusion of phenylacetic acid significantly increased the activity against *E. coli*. Specifically, compound **5a** demonstrated twice the potency of compound **5r**, with the difference attributed to variations in both the aliphatic amine and the acid components. This suggests that the structural modifications in these regions play a crucial role in determining antibacterial efficacy.

Antimicrobial agents are generally classified as either bactericidal or bacteriostatic based on their ability to eliminate pathogens. A drug is considered bactericidal when the ratio of its minimum bactericidal concentration (MBC) to minimum inhibitory concentration (MIC) is low, typically below 4 to 6, and when its concentration is sufficient to kill 99.9% of the targeted organisms [30]. On the other hand, a drug is deemed bacteriostatic when the MBC-MIC ratio is elevated, making it impractical to reach concentrations that can achieve 99.9% bacterial eradication. The distinction between bacteriostatic and bactericidal effects often depends upon the concentration of both the pathogen and the drug within the target tissue. In the case of compounds **5a**–**r**, all have been shown to exhibit bactericidal properties (Figure 8) [5].

#### 2.4.2. Oxidative DNA Damage Studies

Based on the results of MIC and MBC toxicity tests in bacterial models, we decided to modify their bacterial DNA with the compounds analyzed compounds and treat them with the Fpg enzyme (with bifunctional repair glycosylase properties), which serves as an established indicator of oxidative stress (Figure 9 and Appendix A). According to the findings derived from investigations on the modification of bacterial DNA subsequent to digestion with the Fpg protein, we observed that in the *E. coli* R4 strain (ATCC 39543), the structure of the bacterial nucleic acid was destroyed by the formation of the so-called smear on agarose gel. The topological forms of ccc, linear, and oc plasmid DNA were lost in relation to each other. Densely looped structures were formed, creating the so-called concatemer, and their formation may be caused by the accessibility of the enzyme action in its active center to exposed DNA bases, which may be facilitated by the activity of bacterial DNA topoisomerases. (Appendix A). Hence, the Fpg protein encountering such concatemers as a result of modification of bases on guanine and adenine cleaves them, leaving an AP urine site at the site of damage, which is visible on the gel in the form of the analyzed smear. After digestion with the Fpg protein, up to 4% of oxidative damage was identified, indicating that the analyzed compounds very strongly damage bacterial DNA, causing strong oxidative stress in their cell and leading to the disintegration of bacterial DNA (Figure 9).

The aromatic moiety with appended hydroxyl functionalities appears to play a critical role in influencing the toxicity of the *E. coli* R4 strain (ATCC 39543), as demonstrated by the minimum inhibitory concentration (MIC) and minimum bactericidal concentration (MBC) values, which reached statistical significance at *p* < 0.05, as presented in Table 3.

#### 2.4.3. Cytotoxicity Studies

Following the completion of MIC and MBC assays on the analyzed bacterial strains, and subsequent treatment with Fpg protein modified by peptidomimetics along with selected antibiotics, a further MTT assay was conducted to evaluate the cytotoxicity of the compounds employing the BALB/c3T3 mouse embryonic fibroblast cell line (Figure 10).

BALB/c3T3 mouse embryonic fibroblast cells were exposed to peptidomimetics across concentrations ranging from 0.5 µM to 3.5 µM and incubated for a duration of 24 h. Peptidomimetics at the lowest concentration of 0.5 µM exhibited no cytotoxicity to BALB/c3T3 cells, maintaining viability percentages above 99.5%. Nonetheless, a progressive decline in cell viability was observed for all compounds at the concentration of 1.0 µM, with viability percentages varying from 86.9% for compound **5q** to 60.2% for peptidomimetic **5c**. The highest concentrations (2.5 µM and 3.5 µM) inhibited cell viability, which was reduced to 20.2% and 1.0%, respectively (Figure 10 and Appendix A). The results obtained were used to calculate the half maximum inhibitory concentration (IC_50_) after 24 h of incubation with the most active antimicrobial peptidomimetic **5a** (1.61 µM/(±0.14)).

Analogous to the evaluation of compounds **1** and **5a**–**r**, the MTT assay was conducted using representative antibiotics, namely ciprofloxacin (cipro), bleomycin (bleo), and cloxacillin (clox) (refer to Figure 11 and Appendix A). Equivalent concentrations of these antibiotics were employed in the experiments. The findings reveal that the cytotoxicity associated with the tested peptidomimetics **5a**–**r** is either lower than or similar to that of these commonly utilized drugs.

## 3. Materials and Methods

### 3.1. Chemicals

The required reagents were obtained from Sigma-Aldrich (Poznan, Poland) and used as received without any further purification. The water and hexane mixtures were distilled before use, whereas other solvents of analytical grade were used directly without additional drying or purification. The solvents and volatile reagents were evaporated under reduced pressure. All reactions were conducted in desiccated glass vessels under ambient conditions. For the analysis of thin layer chromatography (TLC), silica gel plates 60 F254 from Merck (Poznan, Poland) were used. After evaporation of the solvents, the crude mixtures were purified by column chromatography using Merck silica gel (60/230–400 mesh), with suitable mixtures of hexane and ethyl acetate. Nuclear magnetic resonance (NMR) spectra of ^1^H and ^13^C were recorded in deuterated chloroform (CDCl_3_) and DMSO-*d*_6_ on Bruker 400 MHz and Varian 500 MHz spectrometers, using tetramethylsilane (TMS) as an internal standard. Chemical shifts were expressed in parts per million (ppm) relative to the residual solvent signal, and coupling constants (*J*) were measured in Hertz (Hz). High-resolution mass spectra (HRMS) were obtained on a Maldi SYNAPT G2-S HDMS system equipped with a QqTOF analyzer (Waters, Warsaw, Poland).

#### General Preparation Procedure for Peptidomimetics **5a**–**r**

To a solution of aldehyde **1** (0.5 mmol) in methanol (2.0 mL), the appropriate amine (0.5 mmol) was added, and the mixture was stirred for 10 min. The appropriate carboxylic acid (0.5 mmol) and isocyanide (0.5 mmol) were then added, and the reaction mixture was stirred at 50 °C for 18 h. After being cooled, the solvent was removed under reduced pressure to obtain the crude product. The crude product was purified by flash chromatography using an 8:2 ratio of ethyl acetate to hexanes as an eluent.

***N*-benzyl-2-(3,4-dihydroxyphenyl)-2-(*N*-(4-methoxybenzyl)-2-phenylacetamido)-acetamide** (**5a**). White solid (175 mg, 68%); ^1^H NMR (400 MHz, CDCl_3_) *δ* 7.20–7.13 (m, 7H), 7.09 (d, *J* = 6.7 Hz, 4H), 6.84 (d, *J* = 8.4 Hz, 2H), 6.71–6.65 (m, 3H), 6.55 (d, *J* = 7.1 Hz, 1H), 6.47 (t, *J* = 5.4 Hz, 1H), 5.60 (s, 1H), 5.29 (s, 3H), 4.60–4.35 (m, 2H), 4.26 (d, *J* = 3.4 Hz, 2H), 3.69 (s, 3H), 3.62 (d, *J* = 8.4 Hz, 2H); ^13^C NMR (100 MHz, CDCl_3_) *δ* 173.6, 170.4, 158.8, 145.5, 144.7, 137.5, 134.2, 129.3, 128.8, 128.7, 128.6, 128.5, 127.6, 127.3, 127.3, 126.9, 125.7, 122.4, 116.1, 114.8, 114.0, 63.8, 55.2, 49.9, 43.6, 41.2; HRMS *m*/*z* calcd. for C_31_H_30_N_2_O_5_Na [M + Na]^+^, 533.2050; found: 533.2052; Element. anal. calcd. for C_31_H_30_N_2_O_5_ C 72.92, H 5.92, N 5.49; found: C 72.69, H 6.17, N 5.35.

**2-(3,4-dihydroxyphenyl)-N-(4-methoxybenzyl)-2-(N-(4-methoxybenzyl)-2-phenylacetamido)-acetamide** (**5b**). White solid (130 mg, 67%). ^1^H NMR (400 MHz, DMSO-*d*_6_) *δ* 8.98 (s, 2H), 8.41 (t, *J* = 6.0 Hz, 1H), 7.19 (dt, *J* = 12.8, 5.1 Hz, 3H), 7.10 (d, *J* = 8.2 Hz, 3H), 6.97 (dd, *J* = 30.5, 7.7 Hz, 3H), 6.76 (dd, *J* = 27.7, 7.3 Hz, 5H), 6.57 (q, *J* = 8.4, 7.6 Hz, 2H), 5.92 (s, 1H), 4.60 (d, *J* = 17.2 Hz, 1H), 4.47 (d, *J* = 17.7 Hz, 1H), 4.18 (qd, *J* = 14.9, 5.8 Hz, 2H), 3.68 (s, 3H), 3.65 (s, 3H), 3.38 (d, *J* = 15.6 Hz, 2H).; 13C NMR (100 MHz, DMSO-*d*_6_) δ 172.4, 170.5, 158.5, 158.3, 150.9, 145.6, 141.1, 135.9, 131.6, 131.2, 129.5, 128.9, 128.7, 128.5, 127.5, 126.8, 126.7, 126.6, 121.4, 117.5, 115.7, 114.0, 61.7, 55.4, 55.4, 48.2, 42.0, 40.6; HRMS *m*/*z* calcd. for C_32_H_32_N_2_O_6_Na [M + Na]^+^, 563.2162; found: 563.2158; Element. anal. calcd. for C_32_H_32_N_2_O_6_ calc. C 71.10, H 5.97, N 5.18; found: C 70.85, H 5.98, N 5.30.

**2-(3,4-dihydroxyphenyl)-*N*-(4-fluorobenzyl)-2-(*N*-(4-methoxybenzyl)-2-phenylacetamido)-acetamide** (**5c**). White solid (125 mg, 65%); ^1^H NMR (400 MHz, DMSO-*d*_6_) *δ* 8.48 (d, *J* = 7.8 Hz, 1H), 7.28–7.18 (m, 6H), 7.17–7.12 (m, 1H), 7.05 (dd, *J* = 16.4, 8.0 Hz, 4H), 6.97 (d, *J* = 8.3 Hz, 2H), 6.74 (d, *J* = 9.5 Hz, 2H), 6.56 (t, *J* = 9.0 Hz, 2H), 5.94 (s, 1H), 4.54 (dd, *J* = 47.9, 17.5 Hz, 2H), 4.22 (d, *J* = 15.6 Hz, 3H), 3.51 (ddd, *J* = 16.0, 10.2 Hz, 5H); ^13^C NMR (101 MHz, DMSO-*d*_6_) δ 172.3, 170.7, 161.5 (d, *J* = 242.0 Hz), 158.3, 146.2, 145.9, 136.0, 131.4, 129.6, 128.5, 127.5, 126.7, 126.3, 121.2, 117.6, 115.8, 115.2 (d, *J* = 20.8 Hz), 114.0, 113.3, 61.9, 55.4, 48.2, 41.8, 40.7; HRMS *m*/*z* calcd. for C_31_H_29_FN_2_O_5_Na [M + Na]^+^, 551.1964; found: 551.1958; Element. anal. calcd. for C_31_H_29_FN_2_O_5_ + H_2_O C 68.12, H 5.72, N 5.13; found: C 68.06, H 5.63, N 5.14.

***N*-benzyl-2-(*N*-cyclohexyl-2-phenylacetamido)-2-(3,4-dihydroxyphenyl)acetamide** (**5d**). White solid (100 mg, 59%); ^1^H NMR (400 MHz, Chloroform-d) δ 7.43–7.22 (m, 3H), 7.18 (d, *J* = 8.0 Hz, 2H), 7.13 (d, *J* = 3.2 Hz, 5H), 6.75 (d, *J* = 8.1 Hz, 1H), 6.64 (dd, *J* = 8.2, 2.2 Hz, 1H), 6.51 (t, *J* = 6.0 Hz, 1H), 4.69 (s, 1H), 4.41–4.22 (m, 2H), 3.81–3.71 (m, 2H), 3.64 (d, *J* = 31.1 Hz, 1H), 1.64 (dd, *J* = 68.7, 12.5 Hz, 5H), 1.39–1.30 (m, 1H), 1.23–1.15 (m, 1H), 1.13–0.91 (m, 3H). ^13^C NMR (100 MHz, CDCl_3_) *δ* 172.7, 170.5, 145.4, 144.7, 137.5, 134.5, 128.8, 128.7, 128.5, 127.4, 127.3, 126.9, 126.0, 121.9, 116.2, 115.0, 63.4, 47.3, 43.6, 40.8, 31.6, 19.9, 13.4; HRMS *m*/*z* calcd. for C_29_H_32_N_2_O_4_Na [M + Na]^+^, 495.2261; found: 495.2260. Element. anal. calcd. for C_29_H_32_N_2_O_4_ + H_2_O C 71.0, H 6.99, N 5.71; found: C 71.10, H 6.85, N 5.59.

***N*-cyclohexyl-*N*-(1-(3-dihydroxyphenyl)-2-(4-methoxybenzyl)amino)-2-oxoethyl)-2-phenylacetamide** (**5e**). White solid (100 mg, 55%); ^1^H NMR (400 MHz, DMSO) *δ* 8.77 (s, 3H), 7.30–7.12 (m, 7H), 7.04 (s, 1H), 6.84 (d, *J* = 8.1 Hz, 3H), 6.63 (d, *J* = 10.6 Hz, 1H), 5.40 (s, 1H), 4.36–4.17 (m, 2H), 3.83–3.54 (m, 6H), 1.81–1.36 (m, 6H), 1.23 (m, 2H), 1.06–0.90 (m, 2H). ^13^C NMR (100 MHz, DMSO) δ 169.9, 169.9, 158.7, 158.7, 145.7, 145.1, 137.1, 131.3, 129.9, 129.8, 129.4, 128.9, 128.6, 128.5, 128.2, 127.0, 126.7, 115.8, 114.1, 114.0, 61.5, 60.2, 55.5, 42.3, 41.9, 29.8, 26.4, 26.2, 25.9, 25.6. HRMS calculation. For C_30_H_34_N_2_O_5_Na, [M + Na]^+^, 513.2165, found: 513.2166. Element. anal. for C_30_H_34_N_2_O_5_ calc. C 71.69, H 6.82, N 5.57, found: C 71.45, H 6.72, N 5.29.

***N*-cyclohexyl-*N*-(1-(3-dihydroxyphenyl)-2-(4-fluorobenzyl)amino)-2-oxoethyl)-2-phenyl-acetamide** (**5f**). White solid (85 mg, 48%); ^1^H NMR (400 MHz, DMSO-*d*_6_) *δ* 8.78 (d, *J* = 7.1 Hz, 3H), 7.40–7.10 (m, 9H), 7.06 (s, 3H), 6.54 (dd, *J* = 42.0, 34.0 Hz, 2H), 4.36–4.18 (m, 2H), 3.73 (d, *J* = 4.0 Hz, 2H), 1.64 (d, *J* = 10.6 Hz, 2H), 1.21 (ddd, *J* = 40.5, 11.7, 4.6 Hz, 4H), 1.10–0.98 (m, 2H), 0.97–0.77 (m, 2H). ^13^C NMR (126 MHz, DMSO-*d*_6_) δ 171.2, 170.2, 161.7 (d, *J* = 242.5 Hz), 145.3 (d, *J* = 37.5 Hz), 136.7, 135.9 (d, *J* = 14.7 Hz), 129.7 (d, *J* = 8.0 Hz), 129.5, 129.1, 128.3, 126.5, 120.3, 116.6, 115.8, 115.2 (d, *J* = 21.3 Hz), 62.3, 42.4, 42.1, 30.6, 26.2, 25.4.; HRMS *m*/*z* calcd. for C_29_H_31_FN_2_O_4_Na, [M + Na]^+^, 513.2165; found: 513.2166. Element. anal. calcd. for C_29_H_31_FN_2_O_4_ C 72.62, H 6.77, N 6.27; found: C 72.18, H 7.29, N 6.05.

***N*-benzyl-2-(*N*-butyl-2-phenylacetamido)-2-(3,4-dihydroxyphenyl)acetamide** (**5g**). White solid (100 mg, 62%); ^1^H NMR (400 MHz, CDCl_3_) *δ* 7.21 (d, *J* = 5.6 Hz, 2H), 7.20–7.15 (m, 6H), 7.13–7.09 (m, 2H), 7.06 (d, *J* = 1.8 Hz, 1H), 6.77 (d, *J* = 8.1 Hz, 1H), 6.67 (dd, *J* = 8.2, 1.8 Hz, 1H), 6.50 (t, *J* = 5.8 Hz, 1H), 5.62 (s, 1H), 4.30 (d, *J* = 5.8 Hz, 2H), 3.70 (t, *J* = 10.3 Hz, 2H), 3.29–3.19 (m, 2H), 1.27–1.29 (m, 2H), 1.03–0.98 (m, 2H), 0.68 (t, *J* = 7.1 Hz, 3H); ^13^C NMR (100 MHz, CDCl_3_) *δ* 172.7, 170.5, 145.4, 144.7, 137.5, 134.5, 128.8, 128.7, 128.5, 127.4, 127.3, 126.9, 126.0, 121.9, 116.2, 115.0, 63.4, 47.3, 43.6, 40.8, 31.6, 19.9, 13.4; HRMS calcd. for C_27_H_30_N_2_O_4_Na, [M + Na]^+^ 469.2104; found: 469.2103; Element. anal. calcd. for C_27_H_30_N_2_O_4_ calc. C 72.62, H 6.77, N 6.27, found: C 72.18, H 7.29, N 6.05.

***N*-butyl-*N*-(1-(3,4-dihydroxyphenyl)-2-(4-methoxybenzyl)amino)-2-oxoethyl)-2-phenyl-acetamide** (**5h**). White solid (93 mg, 54%);^1^H NMR (400 MHz, DMSO-*d*_6_) δ 8.94 (s, 2H), 8.31 (dd, *J* = 12.3, 6.4 Hz, 1H), 7.32–7.15 (m, 6H), 7.12 (d, *J* = 8.2 Hz, 2H), 6.83 (dd, *J* = 18.0, 8.1 Hz, 2H), 6.67 (t, *J* = 9.7 Hz, 1H), 6.60–6.50 (m, 1H), 5.83 (s, 1H), 4.21 (td, *J* = 11.9, 9.5, 5.6 Hz, 2H), 3.70 (s, 5H), 3.28–3.03 (m, 2H), 1.38–1.14 (m, 2H), 1.02–0.88 (m, 2H), 0.62 (t, *J* = 7.3 Hz, 3H). ^13^C NMR (100 MHz, DMSO-*d*_6_) *δ* 171.0, 170.4, 158.5, 145.6, 145.5, 136.5, 131.7, 129.7, 129.5, 129.2, 128.9, 128.6, 128.5, 127.1, 126.7, 121.1, 117.5, 115.8, 113.9, 61.0, 55.4, 45.5, 41.9, 40.6, 32.2, 19.9, 13.8; HRMS *m*/*z* calcd. for C_28_H_32_N_2_O_5_Na, [M + Na]^+^, 499.2210; found: 499.2209. Element. anal. calcd. for C_28_H_32_N_2_O_5_.H_2_O calc. C 69.26, H 6.85, N 5.77; found: C 69.05, H 6.80, N 5.49.

***N*-butyl-*N*-(1-(3-dihydroxyphenyl)-2-(4-fluorobenzyl)amino)-2-oxoethyl)-2-phenyl-acetamide** (**5i**). White solid, (95 mg, 56%); ^1^H NMR (400 MHz, DMSO-*d*_6_) *δ* 8.98 (s, 2H), 8.43 (t, *J* = 5.7 Hz, 1H), 7.34–7.21 (m, 8H), 7.09 (t, *J* = 8.7 Hz, 2H), 6.78–6.65 (m, 2H), 5.84 (s, 1H), 4.27 (s, 2H), 3.74 (q, *J* = 15.5 Hz, 2H), 3.30–3.06 (m, 2H), 1.35–1.26 (m, 1H), 1.03–0.89 (m, 2H), 0.79 (d, *J* = 6.6 Hz, 1H), 0.64 (t, *J* = 7.3 Hz, 3H); ^13^C NMR (100 MHz, DMSO-*d*_6_) δ 171.1, 170.6, 161.5 (d, *J* = 241.5 Hz), 145.7, 145.6, 136.3 (d, *J* = 49.9 Hz), 129.6 (d, *J* = 24.2 Hz), 128.6, 127.0, 126.7, 121.2, 117.5, 115.2 (d, *J* = 21.3 Hz), 63.4, 61.1, 45.5, 41.8, 32.2, 19.9, 13.8; HRMS *m*/*z* calcd. for C_27_H_29_FN_2_O_4_Na, [M + Na]^+^, 487.2018, found: 487.2009. Element. anal. calcd. for C_27_H_29_FN_2_O_4_ + 0.3 H_2_O C 69.01, H 6.35, N 5.96; found: C 69.06, H 6.12, N 5.71.

***N*-benzyl-2-(3,4-dihydroxyphenyl)-2-(*N*-(4-methoxybenzyl)acetamido)acetamide** (**5j**). White solid (84 mg, 57%); ^1^H NMR (400 MHz, DMSO-*d*_6_) δ 9.21–8.23 (m, 3H), 7.24 (d, *J* = 7.0 Hz, 2H), 7.21–7.17 (m, 3H), 6.88 (d, *J* = 8.3 Hz, 2H), 6.70 (d, *J* = 8.6 Hz, 2H), 6.62–6.46 (m, 3H), 5.91 (s, 1H), 4.53 (d, *J* = 17.0 Hz, 1H), 4.40 (d, *J* = 17.6 Hz, 1H), 4.25 (d, *J* = 5.8 Hz, 2H), 3.64 (s, 3H), 1.82 (s, 3H). ^13^CNMR (100 MHz, DMSO-*d*_6_) *δ* 172.1, 170.8, 160.0, 158.2, 145.6, 145.4, 139.7, 131.2, 128.7, 128.6, 127.5, 127.5, 127.1, 126.7, 122.9, 121.4, 117.5, 116.7, 115.7, 113.9, 61.6, 55.3, 42.5, 42.4, 22.8, 22.8; HRMS calcd. for C_25_H_26_N_2_O_5_Na, [M + Na]^+^ 457.1740; found: 457.1739. Element. anal. calcd. for C_25_H_26_N_2_O_5_. C 69.11, H 6.03, N 6.45; found: C 69.08, H 6.13, N 6.45.

**2-(3,4-dihydroxyphenyl)-*N*-(4-methoxybenzyl)-2-(*N*-(4-methoxybenzyl)acetamido)-acetamide** (**5k**). White solid (100 mg, 60%); ^1^H NMR (400 MHz, DMSO-*d*_6_) *δ* 8.89 (s, 2H), 8.42 (s, 1H), 7.12 (d, *J* = 8.1 Hz, 2H), 6.90 (d, *J* = 8.2 Hz, 1H), 6.82 (d, *J* = 8.2 Hz, 3H), 6.71 (d, *J* = 8.2 Hz, 3H), 6.57 (t, *J* = 12.5 Hz, 2H), 5.94 (s, 1H), 4.48 (m, 2H), 4.18 (d, *J* = 5.3 Hz, 2H), 3.70 (s, 3H), 3.66 (s, 3H), 1.81 (s, 3H); ^13^C NMR (100 MHz, DMSO-*d*_6_) *δ* 171.8, 170.5, 158.2, 154.9, 145.6, 145.5, 145.4, 131.7, 131.4, 129.2, 128.9, 127.5, 127.0, 121.2, 117.4, 116.8, 115.7, 114.1, 114.0, 113.9, 61.2, 55.5, 55.3, 48.7, 41.9, 22.9; HRMS *m*/*z* calcd. for C_26_H_28_N_2_O_6_Na [M + Na]^+^, 487.1848; found: 487.1845; Element. anal. calcd. for C_26_H_28_N_2_O_6_ C 67.23, H 6.08, N 6.03; found: C 67.08, H 6.01, N 6.13.

**2-(3,4-dihydroxyphenyl)-*N*-(4-fluorobenzyl)-2-(*N*-(4-methoxybenzyl)acetamido)-acetamide** (**5l**). White solid (105 mg, 64%); ^1^H NMR (400 MHz, DMSO-*d*_6_) *δ* 8.88 (s, 2H), 8.49 (s, 1H), 7.23 (d, *J* = 5.4 Hz, 2H), 7.08 (t, *J* = 8.6 Hz, 2H), 6.88 (m, 2H), 6.72 (d, *J* = 7.2 Hz, 2H), 6.67–6.42 (m, 3H), 5.92 (s, 1H), 4.48 (m, 2H), 4.23 (s, 2H), 3.66 (s, 3H), 1.83 (s, 3H). ^13^C NMR (100 MHz, DMSO-*d*_6_) δ 171.8, 170.7, 161.5 (d, *J* = 242.5 Hz), 158.2, 145.6, 145.5, 136.0, 131.3, 129.5 (d, *J* = 6.7 Hz), 128.8, 127.5, 126.9, 121.3, 117.4, 115.8, 115.3 (d, *J* = 20.6 Hz), 113.9, 113.3, 61.5, 55.3, 48.7, 41.8, 22.9.; HRMS *m*/*z* calcd. for C_25_H_25_FN_2_O_5_Na [M + Na]^+^ 475.1650; found: 475.1645; Element. anal. calcd. for C_25_H_25_FN_2_O_5_ C 66.3, H 5.5, N 6.1, found: C 66.2, H 5.5, N 6.1.

***N*-benzyl-2-(*N*-cyclohexylacetamido)-2-(3,4-dihydroxyphenyl)acetamide** (**5m**). White solid (75 mg, 56%); ^1^H NMR (400 MHz, CDCl_3_) *δ* 7.20 (s, 1H), 7.16 (d, *J* = 6.9 Hz, 3H), 7.12–7.01 (m, 3H), 6.81 (d, *J* = 8.1 Hz, 1H), 6.66 (dd, *J* = 8.5, 1.5 Hz, 1H), 6.26 (t, *J* = 5.7 Hz, 1H), 4.70 (s, 1H), 4.46 (dd, *J* = 15.2, 6.6 Hz, 1H), 4.16 (dd, *J* = 14.4, 4.1 Hz, 1H), 3.59 (t, *J* = 11.5 Hz, 1H), 2.09 (s, 3H), 2.01 (d, *J* = 8.9 Hz, 1H), 1.88 (d, *J* = 12.9 Hz, 1H), 1.69 (m, 3H), 1.56–1.50 (m, 1H), 1.37–1.27 (m, 2H), 1.21–1.04 (m, 2H). ^13^C NMR (100 MHz, CDCl_3_) *δ* 172.0, 170., 145.2, 145.1, 138.1, 128.4, 128.3, 127.2, 126.9, 121.1, 114.9, 114.8, 62.8, 60.0, 43.5, 32.2, 31.1, 26.0, 25.7, 25.1, 21.8. HRMS calculation. For C_23_H_28_N_2_O_4_Na [M + Na]^+^, 419.1950, found: 419.1947. Element. anal. for C_23_H_28_N_2_O_4_ calc. C 69.68, H 7.12, N 7.07, found: C 69.58, H 7.35, N 6.95.

**2-(*N*-cyclohexylacetamido)-2-(3,4-dihydroxyphenyl)-*N*-(4-methoxybenzyl)acetamide** (**5n**). White solid (100 mg, 65%); ^1^H NMR (400 MHz, DMSO-*d*_6_) *δ* 8.86 (s, 2H), 8.53 (s, 1H), 7.18 (s, 2H), 6.84 (s, 2H), 6.73–6.58 (m, 2H), 6.49 (s, 1H), 5.26 (s, 1H), 4.23 (m, 2H), 3.70 (s, 4H), 1.79 (m, 3H), 1.57 (m, 6H), 1.32–0.99 (m, 4H). ^13^C NMR (100 MHz, DMSO-*d*_6_) *δ* 171.0, 169.8, 158.7, 145.6, 145.0, 131.4, 129.3, 128.9, 128.8, 120.4, 119.2, 116.7, 115.8, 114.1, 61.7, 58.4, 55.5, 42.3, 31.2, 30.3, 29.8, 26.2, 25.6, 25.0; HRMS *m*/*z* calcd. for C_24_H_30_N_2_O_5_Na [M + Na]^+^, 449.2051, found: 449.2052. Element. anal. calcd. for C_24_H_30_N_2_O_5_ +0.5 H_2_OC 66.19, H 7.17, N 6.43; found: C 66.38, H 7.21, N 6.66.

**2-(*N*-cyclohexylacetamido)-2-(3,4-dihydroxyphenyl)-*N*-(4-fluorobenzyl)acetamide** (**5o**). White solid (100 mg, 66%); ^1^H NMR (500 MHz, DMSO-*d*_6_) δ 8.52 (s, 3H), 7.30 (dd, *J* = 8.3, 5.5 Hz, 2H), 7.07 (t, *J* = 8.7 Hz, 2H), 6.78 (s, 1H), 6.67 (d, *J* = 8.2 Hz, 1H), 6.57 (d, *J* = 8.2 Hz, 1H), 5.17 (s, 1H), 4.30 (d, *J* = 5.7 Hz, 2H), 3.73 (s, 1H), 1.93 (s, 3H), 1.69 (ddd, *J* = 37.2, 27.6, 9.0 Hz, 4H), 1.43 (d, *J* = 81.3 Hz, 2H), 1.25–1.02 (m, 4H). ^13^C NMR (125 MHz, DMSO-*d*_6_) δ 170.6, 170.2, 161.7 (d, *J* = 242.2 Hz), 145.3 (d, *J* = 40.1 Hz), 136.0, 129.7 (d, *J* = 8.1 Hz), 129.1, 119.9, 116.5, 115.8, 115.2 (d, *J* = 21.3 Hz), 62.1, 42.4, 29.3, 26.3, 26.2, 25.5.HRMS *m*/*z* calcd. for C_23_H_27_FN_2_O_4_Na [M + Na]^+^ 437.1851; found: 437.1853; Element. anal. calcd. for C_23_H_27_FN_2_O_4_ C 66.65, H 6.57, N 6.76, found: C 66.58, H 6.48, N 6.73.

***N*-benzyl-2-(*N*-butylacetamido)-2-(3,4-dihydroxyphenyl)acetamide** (**5p**). White solid (80 mg, 63%);^1^H NMR (400 MHz, DMSO-*d*_6_) δ 8.93 (d, *J* = 21.5 Hz, 2H), 8.38 (t, *J* = 6.0 Hz, 1H), 7.29–7.18 (m, 5H), 6.75–6.63 (m, 2H), 6.56–6.49 (m, 1H), 5.83 (s, 1H), 4.28 (dd, *J* = 22.4, 5.8 Hz, 2H), 3.17 (dtd, *J* = 26.2, 15.1, 4.7 Hz, 2H), 2.01 (d, *J* = 18.1 Hz, 3H), 1.31–1.20 (m, 1H), 0.96 (qq, *J* = 13.5, 6.9 Hz, 2H), 0.82–0.69 (m, 1H), 0.64 (dd, *J* = 8.1, 6.7 Hz, 3H).; ^13^C NMR (100 MHz, DMSO-*d*_6_) δ 170.6, 170.5, 145.6, 145.6, 139.9, 128.7, 128.5, 127.8, 127.5, 127.3, 127.0, 121.1, 117.4, 115.8, 60.7, 45.9, 42.5, 31.9, 22.1, 19.9, 13.8; HRMS *m*/*z* calcd. for C_21_H_26_N_2_O_4_Na [M + Na]^+^, 393.1791; found: 393.1790. Element. anal. calcd. for C_21_H_26_N_2_O_4_ C 68.09, H 7.07, N 7.56, found: C 67.83, H 6.92, N 7.54.

**2-(*N*-butylacetamido)-2-(3,4-dihydroxyphenyl)-*N*-(4-methoxybenzyl)acetamide** (**5q**). White solid (85 mg, 59%); ^1^H NMR (400 MHz, DMSO-*d*_6_) *δ* 8.92 (s, 2H), 8.31 (t, *J* = 5.6 Hz, 1H), 7.12 (d, *J* = 8.3 Hz, 2H), 6.82 (d, *J* = 8.4 Hz, 2H), 6.72–6.63 (m, 2H), 6.51 (d, *J* = 7.7 Hz, 1H), 5.82 (s, 1H), 4.20 (t, *J* = 11.3 Hz, 2H), 3.70 (s, 3H), 3.21–3.02 (m, 2H), 2.03 (s, 3H), 1.38–1.09 (m, 2H), 1.01–0.86 (m, 2H), 0.64 (t, *J* = 7.3 Hz, 3H); ^13^C NMR (100 MHz, DMSO-*d*_6_) *δ* 170.4, 170.4, 158.5, 145.5, 131.8, 129.2, 128.9, 127.4, 121.0, 117.4, 116.6, 115.8, 114.1, 114.0, 60.6, 55.5, 45.9, 41.9, 31.8, 22.1, 19.9, 13.8; HRMS *m*/*z* calcd. for C_22_H_28_N_2_O_5_Na [M + Na]^+^ 423.1897; found: 423.1896. Element. anal. calcd. for C_22_H_28_N_2_O_5_ C 65.98, H 7.05, N 7.00; found: C 65.74, H 6.97, N 6.78.

**2-(*N*-butylacetamido)-2-(3,4-dihydroxyphenyl)-*N*-(4-fluorobenzyl)acetamide** (**5r**). White solid (85 mg, 60%); ^1^H NMR (400 MHz, DMSO-*d*_6_) *δ* 8.94 (s, 2H), 8.39 (t, *J* = 5.8 Hz, 1H), 7.29–7.21 (m, 2H), 7.09 (dd, *J* = 16.7, 7.9 Hz, 2H), 6.74–6.64 (m, 2H), 6.51 (d, *J* = 7.9 Hz, 1H), 5.80 (s, 1H), 4.25 (t, *J* = 11.7 Hz, 2H), 3.23–3.08 (m, 2H), 2.03 (s, 3H), 1.32–1.18 (m, 1H), 1.03–0.88 (m, 2H), 0.76 (dd, *J* = 12.4, 6.2 Hz, 1H), 0.63 (t, *J* = 7.3 Hz, 3H); ^13^C NMR (100 MHz, DMSO-*d*_6_) δ 170.6, 170.5, 161.5 (d, *J* = 241.7 Hz), 145.6, 135.9 (d, *J* = 30.8 Hz), 129.8, 129.5 (d, *J* = 8.2 Hz), 127.2, 121.1, 120.3, 117.4, 116.6, 115.8, 115.2 (d, *J* = 20.9 Hz), 60.8, 45.9, 41.8, 31.8, 22.1, 19.9, 13.82; HRMS *m*/*z* calcd. for C_21_H_25_FN_2_O_4_Na [M + Na]^+^, 411.1695, found: 411.1696. Element. anal. calcd. for C_21_H_25_FN_2_O_4_ C 64.93, H 6.49, N 7.21, found: C 65.02, H 6.65, N 7.06.

### 3.2. Microorganisms and Media

The reference bacterial strains of *Escherichia coli* (K12 ATCC 25404, R2 ATCC 39544, R3 ATCC 11775, and R4 ATCC 39543), *Staphylococcus aureus* (ATCC 23235), *Acinetobacter baumannii* (ATCC 17978), *Pseudomonas aeruginosa* (ATCC 15442), and *Enterobacter cloacae* (ATCC 49141) were procured from LGC Standards, Teddington, UK, and utilized in compliance with ISO 11133 guidelines [31]. These strains served to assess the antibacterial efficacy of the compounds by determining their minimum inhibitory concentration (MIC) and minimum bactericidal concentration (MBC) in accordance with the procedure outlined in the literature [32].

### 3.3. Tyrosinase Assay

Tyrosinase (Tyr) activity was measured using L-DOPA as substrate and phosphate buffer solution (PBS) as blank control, following the method described by Min He et al. (2022) with an ultraviolet spectrophotometer (Carry-300, Agilent Life Sciences and Chemical Analysis, Santa Clara, CA, USA) [33]. The substrate L-DOPA (0.5 mM) and Tyr solution (200 U/mL) were prepared in PBS, while the solutions of the compounds tested or kojic acid (KA) were dissolved in DMSO and diluted with PBS, ensuring that the final concentration of DMSO concentration was less than 0.1%. Before the experiment, 2.9 mL of PBS and 0.1 mL of DMSO were used to account for the influence of the background solvent. The inhibitor solution (100 µL), substrate solution (2.8 mL), and Tyr solution (100 µL) were then added to a quartz colorimetric cuvette. After a one-minute mixing period, the absorption spectra of the mixture were continuously recorded at 475 nm using the UV spectrophotometer (Carry-300) for 5 min over a period of every 30 s. The tyrosinase inhibition rate was calculated as follows:Relative enzyme activity (%) = (A_1_ − A_2_)/A_1_ × 100

A_1_ = Absorbance of L-DOPA and inhibitor.

A_2_ = Absorbance of L-DOPA, inhibitor, and tyrosinase.

In this assay, kojic acid was used as a positive control, and the same biological sample was repeated three times.

### 3.4. Minimum Inhibitory Concentration (MIC) and Minimum Bactericidal Concentration (MBC)

Minimum inhibitory concentration (MIC) and minimum bactericidal concentration (MBC) were ascertained through the employment of a microtiter plate technique within sterile 48 or 96-well plates, following the published protocol [22]. In summary, MIC and MBC, defined as the lowest concentration at which bacterial growth is inhibited, were evaluated. Initially, 50 μL of the test compounds and corresponding bacterial strains were introduced into the first row of the microtiter plate. Sterile Tryptone Soya Broth (TSB) medium (25 μL) was then added to the following wells, and serial dilutions were executed. Thereafter, 200 μL of inoculated TSB medium, with resazurin (0.02 mg/mL) serving as a redox indicator, was added to each well. The bacterial inoculum was standardized to 10^6^ colony-forming units (CFU)/mL, equivalent to approximately 0.5 McFarland units. The plates were maintained at 30 °C for an incubation period of 24 h. A shift in color from blue to either pink or yellow, accompanied by turbidity, was interpreted as an indicator of bacterial proliferation. The minimal concentration at which a lack of visible color alteration was detected was documented as the MIC [22]. Each determination of MIC and MBC was repeated at least three times. For the estimation of MBC, dehydrogenase activity was measured by observing the color change in triphenyl tetrazolium chloride (TTC) to triphenyl formazan (TF). A dense bacterial culture (approximately 10^7^ CFU/mL) was prepared by incubating the strains in TSB medium at 25 °C for 24 h and then aliquoting them into test tubes. A stock solution of the test compounds was formulated in DMSO at a concentration of 1 mM. The tests included both solvent and negative controls, along with a positive control (a known antibacterial agent with established MIC against the bacterial strains). According to the latest guidelines of the Clinical and Laboratory Standards Institute (CLSI), when carbapenemase-producing enteric bacilli producing carbapenemase are detected, the MIC values of carbapenems must be determined, a class of β-lactam antibiotics structurally distinct -lactam antibiotics from penicillins and cephalosporins. Test compounds were added to the tubes, achieving final concentrations ranging from 10 to 250 mg/mL. The cultures underwent incubation at 30 °C for a duration of 1 h. Subsequently, the test tubes were sealed using parafilm and subjected to further incubation at 30 °C for an additional hour in the absence of light. The MBC was documented as the minimal concentration at which the absence of red coloration (formazan) was discerned.

### 3.5. MTT Assay

The cytotoxic effects of aldehyde 1 and peptidomimetics **5a**–**r** on BALB/c3T3 mouse fibroblast cells were assessed utilizing the MTT assay following a 24 h incubation period across five different concentrations (0.5, 1.0, 1.5, 2.5, and 3.5 µM). The MTT assay is predicated upon the capacity of mitochondrial dehydrogenase enzymes to convert the water-soluble tetrazolium salt, 3-(4-dimethylthiazol-2-yl)-2,5-diphenyltetrazolium bromide (MTT), into an insoluble formazan product, manifesting as dark blue crystals. These formazan crystals are then dissolved in DMSO or isopropanol, forming a colored solution whose intensity is measured spectrophotometrically within the wavelength range of 492–570 nm. The amount of MTT reduced corresponds to the oxidative activity of the mitochondria, which is directly proportional to the number of metabolically active cells in the population under defined experimental conditions. The MTT assay can also be used to assess cell viability in non-dividing metabolically active cells. It is one of the most widely used methods to assess cytotoxic activity and is recommended by international standards organizations as a reference method [34,35].

### 3.6. Estimation of Oxidized Damage Based on Bacterial DNA Digestion by the Fpg Protein

In light of the MIC values, the *E. coli* R4 strain (ATCC 39543) was chosen for advanced DNA analysis. The 16 compounds under examination were identified as disruptors of DNA structure, even following digestion with the Fpg protein (formamidopyrimidine DNA N-glycosylase/AP lyase) in vitro over a 24 h period. Digestion of DNA by the Fpg protein in both control and peptidomimetic-treated cultures exhibited discernible DNA damage, characterized by modifications in the covalently closed circle (ccc), linear, and open circle (oc) forms, as well as the emergence of indistinct bands or ‘smears’, as illustrated in Appendix A. Plasmid samples of *E. coli* R4 (ATCC 39543) primarily presented three forms: oc (very poor), linear, and ccc. Noteworthy differences between control and peptidomimetic-altered plasmids were observed in the electrophoretic images (Appendix A). For this experiment, bacterial DNA was isolated from 2 mL of fresh culture utilizing the New England Biolabs Kit (Labjot, Warsaw, Poland) as per the manufacturer’s instructions. The isolated DNA underwent treatment with 1 mg/mL of each synthesized peptidomimetic. The Fpg protein (New England Biolabs, cat. no. M0240S, 8000 U/mL) was diluted 50 times with 10× NEB buffer, combined with a 100× BSA solution (both provided by the manufacturer of the Fpg protein), and subsequently incubated with 8 µL of purified bacterial DNA, 2 µL of Fpg solution, and 2 µL of NE buffer at 37 °C for 30 min. Control samples (devoid of tested compounds), along with digested and undigested genomic DNA, were assessed via 1% agarose gel electrophoresis. DNA concentration was determined spectrophotometrically through the A260/A280 ratio, while the extent of oxidative damage was quantified using ImageQuant TL 10.2 software.

### 3.7. Statistical Analysis

All experimental data, derived from a minimum of three independent trials (n = 3), are presented as means ± standard error of the mean (SEM; sourced from the manufacturer, Saint Louis, MO, USA). Statistical comparisons between pairs of means were performed using Tukey’s post hoc test. The levels of statistical significance were designated as * *p* < 0.05, ** *p* < 0.1, and *** *p* < 0.01 [36].

### 3.8. Molecular Docking

Molecular docking was performed using the crystal structure of the target protein obtained from the Protein Data Bank (PDB ID: 2Y9X). The protein was prepared by removing water molecules, ligands, and co-factors, followed by the addition of hydrogen atoms and energy minimization using the discovery studio visualizer (v24.1.0.23298). The binding site was defined around the active site. Synthesized peptidomimetics (compounds **1**–**19**) were modeled using ChemDraw (version 23.1.2.7), energy-minimized with the MMFF94 force field, and converted into appropriate formats for docking. Docking simulations were conducted using PyRx (version 0.8) with AutoDock Vina (version 1.2.0), with a grid box centered on the active site. The Lamarckian Genetic Algorithm (LGA) was employed, and the exhaustiveness parameter was set to 8 for thorough conformational exploration.

## 4. Conclusions

In this study, we successfully synthesized a series of peptidomimetics, with the aim of combining their dual functionalities as tyrosinase inhibitors and antimicrobial agents. By employing 3,4-dihydroxybenzaldehyde as the core structure in synthesis, we designed peptidomimetics that mimic the structure of L-DOPA, a known substrate of tyrosinase. The resulting compounds exhibited significant tyrosinase inhibitory activity with micromolar values, demonstrating their potential as candidates for depigmentation therapies. During the biological activity testing, the peptidomimetics showed broad-spectrum antimicrobial activity, particularly against *E. coli* and other Gram-negative bacteria, surpassing the efficacy of known antibiotics in some cases. This dual functionality highlights the versatility of these compounds for use in both cosmetic and pharmaceutical applications, including the prevention of enzymatic browning in food and the treatment of microbial infections. The cytotoxicity of the synthesized compounds was assessed employing the MTT assay on BALB/c3T3 mouse fibroblast cells, which verified that the peptidomimetics demonstrate low toxicity at therapeutic concentrations, analogous to existing antibiotics. In particular, peptidomimetic **5a** showed double the potency of compound **5r**, illustrating the critical role of both the aromatic amine and the acid components in enhancing antimicrobial efficacy. Furthermore, our results indicate that the peptidomimetics interfere with bacterial DNA structure, as evidenced by the Fpg digestion assay, further supporting their potential as novel antimicrobial agents. In general, this research contributes to the growing field of dual-activity therapeutics by providing new insights into the design of multifunctional peptidomimetics. The ability of these compounds to inhibit tyrosinase and combat antimicrobial resistance simultaneously could significantly reduce the cost and time of drug discovery, offering promising solutions for food preservation and clinical challenges with the treatment of infected skin wounds. Future studies will explore structural optimizations to enhance their potency and selectivity, particularly against drug-resistant bacterial strains.

## Data Availability

This is available at the request of those interested.

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
