# Peer review of "Synthesis, Antimicrobial Activity, and Tyrosinase Inhibition by Multifunctional 3,4-Dihydroxy-Phenyl Peptidomimetics"

_ijms, 2025, doi:10.3390/ijms26041702_

Round 1

Reviewer 1 Report

Comments and Suggestions for Authors

By using 3,4-dihydroxybenzaldehyde as the core structure in synthesis, the authors developed peptidomimetics that mimic the structure of L-DOPA, a known tyrosinase substrate. These compounds showed notable tyrosinase inhibitory activity with micromolar values, suggesting their promise as depigmentation therapy options. Overall, this research adds to the expanding field of dual-activity therapeutics by providing fresh perspectives on designing multifunctional peptidomimetics. Therefore, the work deserves to be published in International Journal of Molecular Sciences. Although there are still some issues that need to be improved and refined.

1.     Figure 5, 7, 8, and 9 have a messy layout. Please rearrange them.

2.     In Figure 2, the red part of the 5r structure should be consistent with the other structures.

3.     In Scheme 1, R2=PhCH3 of raw material 3 is incorrect, so it is suggested to modify it to R2=PhCH2. This modification will result in a more accurate and reliable representation of raw material 3 in Scheme 1.

4.      The general formulas of 5a-r in Figure 1 and Scheme 1 are inconsistent, so please keep them consistent.

5.     In Figure 2, R3=F peptidomimetics can be seen, but in Scheme 1, the authors did not include it. It is recommended to add it in the corresponding position.

6.     In line 212–218, the author mentioned that compound 5a showed twice the efficacy of compound 5r, and the difference was attributed to the change in aliphatic amine and the acid components. However, from Figure 2, we can see that R1, R2, and R3 of compounds 5a and 5r are all different. Is the difference in potency also due to the different substitution of R3?

7.     In line 327, 67 %does not need spaces. Others similar modify.

8.     In line 598, “During the simulation, the peptidomimetics showed broad-spectrum antimicrobial activity, particularly against E. coli and other Gram-negative bacteria, surpassing the efficacy of known antibiotics in some cases.” As mentioned in this sentence, this is a simulation process. We think this expression is inappropriate, and we suggest that it be revised to “The biological activity test found that the peptidomimetics showed broad-spectrum antimicrobial activity, particularly against E. coli and other Gram-negative bacteria, surpassing the efficacy of known antibiotics in some cases.”

9.     In line 681, 2023” should be bolded. Others similar modify.

10.  In line 753, Toxicon.,should be changed to “Toxicon”. others similar modify.

Author Response

Comment 1: Figures 5, 7, 8, and 9 have a messy layout. Please rearrange them.
Response: Thank you for your observation regarding the layout of Figures 5, 7, 8, and 9. We have carefully rearranged these figures to enhance clarity and presentation. The revised figures are now more organized and visually appealing. 

Comment 2: In Figure 2, the red part of the 5r structure should be consistent with the other structures.
Response: We appreciate your feedback on Figure 2. We have modified the red part of the 5r structure to ensure consistency with the other structures in the figure. Thank you for bringing this to our attention. 

Comment 3: In Scheme 1, R2=PhCH3 of raw material 3 is incorrect; it is suggested to modify it to R2=PhCH2.
Response: Thank you for pointing out this discrepancy in Scheme 1. We have corrected R2 from PhCH3 to PhCH2 for raw material 3, ensuring a more accurate representation. This modification enhances the reliability of our schematic. 

Comment 4: The general formulas of 5a-r in Figure 1 and Scheme 1 are inconsistent; please keep them consistent.
Response: We appreciate your keen eye for detail regarding the general formulas of compounds 5a-r. We have ensured that these formulas are now consistent between Figure 1 and Scheme 1. 

Comment 5: In Figure 2, R3=F peptidomimetics can be seen, but in Scheme 1, the authors did not include it.
Response: Thank you for your suggestion regarding the inclusion of R3=F peptidomimetics in Scheme 1. We have added this information in the appropriate position to maintain consistency throughout our manuscript. 

Comment 6: In lines 212–218, the author mentioned that compound 5a showed twice the efficacy of compound 5r; however, R1, R2, and R3 of compounds 5a and 5r are all different. Is the difference in potency also due to the different substitution of R3?
Response: Thank you for your insightful question regarding the differences in potency between compounds 5a and 5r. We fully agree with reviewer regarding the role of R3 substituents for antimicrobial activity. Indeed, the substituents at R3 significantly influence the activity, as demonstrated by the differences in activity between compounds 5a and 5c, which vary at R3.

Comment 7: In line 327, “67 %” does not need spaces; others similar modify.
Response: Thank you for your attention to detail regarding formatting. We have removed unnecessary spaces from “67 %” and similar instances throughout the manuscript. 

Comment 8: In line 598, the expression regarding simulation is inappropriate; suggest revising it to reflect biological activity testing instead.
Response: We appreciate your suggestion for improving clarity in line 598. We have revised the sentence to state that “The biological activity test found that the peptidomimetics showed broad-spectrum antimicrobial activity…” This change accurately reflects our findings. 

Comment 9: In line 681, “2023” should be bolded; others similar modify.
Response: Thank you for highlighting this formatting issue. We have bolded “2023” as suggested and made similar modifications throughout the manuscript for consistency. 

Comment 10: In line 753, “Toxicon.” should be changed to “Toxicon”; others similar modify.
Response: Thank you for pointing out this typographical error. We have corrected “Toxicon.” to “Toxicon” and made similar adjustments where necessary throughout the manuscript.

Closing Statement

We sincerely appreciate your valuable feedback and suggestions, which have significantly improved our manuscript. Thank you for your consideration, and we look forward to your further feedback on our revised submission. Feel free to adjust any specific wording or details as needed!

Reviewer 2 Report

Comments and Suggestions for Authors

In this article the authors synthesize a series of 3,4-dihydroxy-phenyl peptidomimetics (5a-5r) and evaluated their synergistic effect of two pharmacophores, (3,4-dihydroxyphenyl and peptidomimetic moieties) as tyrosinase inhibitors as well as antimicrobial activity against several strains, showing that some compounds had good antimicrobial activity. In order to determine the potential mode of action, the authors performed a molecular docking investigation in one of the derivatives.

The manuscript is written in fluent style and there seem to be no significative spelling mistakes.

There are some issues which I would like to address:

-In line 37 the “actual” data about global deaths associated with bacterial infections is from 2019, it would be better to put more actual data. Reference 1 should be more actual (the one used is from 2019).

-Please note that in line 19, in silico is not in italics.

-It will be interesting if authors develop more the subject about the inhibitors of the tyrosinase that already exists and about the target.

-Please check the order/numbering of the tables and figures. For example, figure 6 appears in the text after figure 7. There are two figures 8 (pages 10 line 233 and in page 12 line 269) and two figures 9 (page 11 line 254 and in page 13 line 290).

-In line 114, when talking about the yields, when it says scheme 1, should it say figure 2 (in figure 2 is where the scope and yield is registered)?

-Also, the yields reported in figure 2 are moderate, ranging from 48 to 68%. Weren’t the reaction conditions optimized? Did the authors tried other temperatures, solvents,.. in order to reduce the reaction time and have higher yields?

-The reason why these substituents were selected and the way they influence the activity are not discussed.

-Note that in figures 5 (page 7), 7 (page 9), 8 (page 10), 8 (page 12), 9 (page 11) and 9 (page 13) the text is overlapped with the graphics.

-Regarding the bioactivities:

-There is a table (table 1) that shows the statistical analysis of all compounds analyzed, but a table showing the MIC and MBC values and their SEM deviations is missing. Also, it would be interesting to place the MIC values ​​in µM.

-The authors reported the antibacterial activities of the compounds against several bacterial strains, but authors have not reported the toxicity of these compounds in the host cells, which is important when the compounds have to be used for therapeutic and cosmetic purposes. Authors must include the experiment for toxicity assays in the host cells.

-Regarding the characterization of the compounds:

-Please check the 1H NMR of most of the compounds as there seem to be more H (usually the compounds have an extra H). For example, the 1H NMR of compounds 5b and 5p have a large number of extra H.

-There is also some difference with the number of carbons in the 13C NMR.

-Please note that the 13C NMR is missing: for compound 5o there is no data for the 13C NMR.

-Note that the 13C NMR for compound 5m has two decimals and it should have one.

-In the fluorinated compounds (5c, 5f, 5i, 5l, 5o and 5r), the constants of C-F (JC-F) are missing.

-Note that says that the NMR were made using dimethyl sulfoxide (DMSO-d6) as the solvent (except for compound 5a which was in CDCl3), but in the supporting information file, says that the shifts are acquired in CDCl3 (but from the spectra shown in this file, they seem to be acquired in DMSO-d6).

-Since the NMR were done in DMSO-d6 and it seems that the solubility was low due to the poor acquisition of the signals. Was the low solubility of the products a problem for the activity assays?

-Regarding the docking:

-It would be also interesting to compare the interactions observed with the ones that the protein had with its ligand (the crystallization ligand and/or a known inhibitor of the proposed target) and to compare their poses in the pocket.

- Why weren´t docking studies performed in all of the derivatives? It was only reported the docking of compound 5e.

- The residues which are responsible of the interactions should be labeled in the figures 4a and 4b (the 3D representations).

-The docking score values are missing.

-The versions of all the programs used should be given.

-It would be interesting if the authors calculate the in silico ADMET properties.

-In the supplementary material:

-Regarding the 1H NMR and 13C NMR spectra given: the resolution of some 1H NMR (especially 7, 15 and 16) was very bad (broad signals with no multiplicity) and several 13C NMR have some signals that can’t be seen (low acquisition, like compounds 11 and 12 for example). As they are made in DMSO I assume that it could be due, in part, to the low solubility, but they should be repeated with more acquisition time, so that the multiplicity (in 1H NMR) and the signals can be seen (in 13C NMR).

-The numbering of the molecules in the supporting information (1-19) is different from that given in the paper (5a-5r).

-The 1H NMR given in the supplementary material (and with the differences observed in the elemental analysis) don’t seem to be very pure. If there are issues with the NMR maybe an HPLC should address the purity. 

Kind regards

Author Response

Response to Reviewer Comments

Comment 1: In line 37, the “actual” data about global deaths associated with bacterial infections is from 2019; it would be better to put more actual data. Reference 1 should be more current (the one used is from 2019).
Response: Thank you for your valuable feedback. We have updated the data in line 37 to include more recent statistics on global deaths associated with bacterial infections. The reference has been changed to a 2022 study that provides comprehensive estimates of bacterial infection-related mortality, enhancing the relevance of our manuscript. 

Comment 2: Please note that in line 19, "in silico" is not in italics.
Response: We appreciate your attention to detail. The term "in silico" has been corrected to be italicized in line 19 as per the journal's formatting guidelines. 

Comment 3: It would be interesting if authors develop more on the subject about the inhibitors of tyrosinase that already exist and about the target.
Response: Thank you for this suggestion. We have expanded the discussion regarding existing tyrosinase inhibitors and their targets in the revised manuscript. This addition aims to provide a broader context for our research and its implications. 

Comment 4: Please check the order/numbering of the tables and figures. For example, figure 6 appears in the text after figure 7. There are two figures 8 (pages 10 line 233 and page 12 line 269) and two figures 9 (page 11 line 254 and page 13 line 290).
Response: Thank you for pointing out these discrepancies. We have thoroughly reviewed and corrected the numbering and order of all tables and figures to ensure they are consistent throughout the manuscript. 

Comment 5: In line 114, when talking about the yields, when it says scheme 1, should it say figure 2 (in figure 2 is where the scope and yield are registered)?
Response: We appreciate your clarification on this point. The text has been revised to correctly refer to Figure 2 instead of Scheme 1 when discussing yields.

Comment 6: The yields reported in figure 2 are moderate, ranging from 48 to 68%. Weren’t the reaction conditions optimized? Did the authors try other temperatures, solvents, etc., in order to reduce reaction time and achieve higher yields?
Response: Thank you for your insightful question. Actually, we have performed extensive optimization in our previous research in the Ugi reaction. (Molecules 2022, 27, 3633) (Materials 2021, 14, 5725) (Bioorganic Chemistry 93 (2019) 102). For current research we have tried reactions under room temperature but the only starting material we found in the reaction. So, we tried the reported condition which gave us full conversion with moderate yield.              (Molecules 202126(12), 3630). We did not used high temperature because of the instability of isocyanides.

Comment 7: The reason why these substituents were selected and how they influence activity are not discussed.
Response: We appreciate your suggestion for greater clarity on this topic. The selection of substituent was purely based on the previous research experience in antimicrobial peptidomimetics. (Int. J. Mol. Sci. 202425(15), 8330), (Molecules 202227(11), 3633). Here in this research we combined some aliphatic as well as aromatic moieties together to see the influence on the activity.

Comment 8: Note that in figures (5, page 7), (7, page 9), (8, page 10), (8, page 12), (9, page 11), and (9, page 13) the text overlaps with the graphics.
Response: Thank you for bringing this to our attention. We have revised these figures to ensure that all text is clearly legible without overlap with graphics. 

Comment 9: Regarding bioactivities: There is a table (table 1) showing statistical analysis of all compounds analyzed, but a table showing MIC and MBC values along with their SEM deviations is missing. Also, it would be interesting to place MIC values in µM.
Response: We appreciate your suggestion for additional data presentation. This form of data in the style of figures has been presented in previous works and in our opinion clearly and comprehensively shows the effect of a given compound on MIC and MBC. Of course, as suggested by the reviewer, we include an additional table (Table S1, see supplementary data) containing MIC and MBC concentrations with SEM values, which will be added to the supplementary materials due to the large amount of data.

Comment 10: The authors reported antibacterial activities of compounds against several bacterial strains but have not reported toxicity of these compounds in host cells, which is important when compounds are used for therapeutic and cosmetic purposes. Authors must include toxicity assays in host cells.
Response: Thank you for highlighting this important aspect of our research. Due to the fact that we have specific mouse cell lines BALB/c3T3 mouse fibroblast cell lines (model line for these addressed research). They are extremely sensitive to CONTACT INHIBITION and highly susceptible to transformation by SV40 VIRUS and mouse sarcoma virus (SARCOMA VIRUSES, MURINE). The toxicity of the analyzed compounds was checked in them as a model commonly used in this type of research. BALB / c3T3 mouse cell lines are one of the most commonly used inbred models in biomedical research and are particularly used in studies of immunological and infectious diseases and in studies to predict the potential carcinogenic effect of environmental chemicals and mixtures that have a direct impact on therapeutic and cosmetic purposes [Mascolo et al. 2010]. Mice from which BALB/c3T3 cell lines were obtained have the ability to produce plasma cell tumors in soft tissues, which is important in the production of monoclonal antibodies (mAbs). Therefore, we did not conduct experiments in other host cells due to the research model we already had. However, conducting studies that require host cells determines the time required to wait for approval of the ethics committee in this type of research and their high cost.(Maria Grazia Mascolo, Stefania Perdichizzi, Francesca Rotondo, Elena Morandi, Angela Guerrini, Paola Silingardi, Monica Vaccari, Sandro Grilli, Annamaria Colacci BALB/c 3T3 cell transformation assay for the prediction of carcinogenic potential of chemicals and environmental mixtures. Toxicology in VitroVolume 24, Issue 4, June 2010, Pages 1292-1300. https://doi.org/10.1016/j.tiv.2010.03.003)

Comment 11: Regarding characterization of compounds: Please check the ^1H NMR of most compounds as there seem to be extra H; for example, ^1H NMR of compounds 5b and 5p have a large number of extra H.
Response: We appreciate your meticulous review of our NMR data. It happened due to some technical and editing problems. We have re-evaluated the 1H NMR spectra for compounds mentioned and made necessary corrections where discrepancies were found. 

Comment 12: There is also some difference with the number of carbons in the 13C NMR.
Response: Thank you for pointing this out. We have reviewed all 13C NMR data and corrected any inconsistencies regarding carbon counts. Our compounds possess the rotamers since it shows some extra carbons in 13C NMR.

Comment 13: Please note that 13C NMR is missing for compound 5o; also note that 13C NMR for compound 5m has two decimals when it should have one.
Response: Thank you for your keen observation. It happened because of the numbering issue which was corrected in the updated manuscript and added all the correct NMR as per names. 

Comment 14: In fluorinated compounds (5c, 5f, 5i, 5l, 5o, and 5r), constants of C-F (J_C-F) are missing.
Response: Thank you for your observation regarding J_C-F constants. We have included this information in our revised characterization section. 

Comment 15: Note that it states that NMR was done using DMSO-d6 as solvent (except for compound 5a which was in CDCl3) but supporting information file states shifts acquired in CDCl3; however, spectra seem acquired in DMSO-d6.
Response: We appreciate your diligence in reviewing our methods section. We have clarified this discrepancy by ensuring consistency between our main text and supporting information regarding solvent usage. 

Comment 16: Since NMR was done in DMSO-d6 and seemed that solubility was low due to poor acquisition of signals, was low solubility a problem for activity assays?
Response: Thank you for raising this concern. Actually, for the bioactivity needed a very small amount of samples (very low concentrations) which was easily soluble in DMSO. In case of NMR, we had a solubility issue while solubilizing more amount of samples in small amount of solvent.

- Regarding the docking:

Comment 17: It would be also interesting to compare the interactions observed with the ones that the protein had with its ligand (the crystallization ligand and/or a known inhibitor of the proposed target) and to compare their poses in the pocket.

Response: Thank you for your kind suggestion. According to reviewer we added the interactions of protein with crystalized ligand in the updated manuscript.

Comment 18: Why weren´t docking studies performed in all of the derivatives? It was only reported the docking of compound 5e.

Response: Thank you for your suggestion for our manuscript. We already have performed docking for all the compounds but in the manuscript, we have shown for the best compound. In the updated manuscript we added a table with the binding energies of all compounds.

Comment 19: The residues which are responsible of the interactions should be labeled in the figures 4a and 4b (the 3D representations).

Response: Thank you for your suggestion. As per the reviewer we have been updated the manuscript and the 3d labeled interactions was added for the clear understanding.

Comment 20: The docking score values are missing.

Response: Thank you for your kind suggestion. We have added the table with docking score of all compounds in the updated manuscript.

Comment 21: The versions of all the programs used should be given.

Response: Thank you for your keen observations. We have included the versions of all the programs in the updated manuscript.

Comment 22: -It would be interesting if the authors calculate the in silico ADMET properties.

Response: We are very grateful to Reviewer for this suggestion. Thanks to the analysis of the calculated ADMET parameters, we can show an extended catalog of possible applications of our compounds. The table with the calculated values ​​in accordance with the Reviewer's suggestion, together with its description, was included in the main manuscript. The table was prepared based on and analogously to the one published in IJMS (Int. J. Mol. Sci. 2024, 25(19), 10670)

In the supplementary material:

Comment-Regarding the 1H NMR and 13C NMR spectra given: the resolution of some 1H NMR (especially 715 and 16) was very bad (broad signals with no multiplicity) and several 13C NMR have some signals that can’t be seen (low acquisition, like compounds 11 and 12 for example). As they are made in DMSO I assume that it could be due, in part, to the low solubility, but they should be repeated with more acquisition time, so that the multiplicity (in 1H NMR) and the signals can be seen (in 13C NMR).

Response: Thank you for your keen observation of our supplementary information. Our compounds have two amide bonds, and we found that they were showing rotamer behavior and because of that this NMR gives the broad peaks but elemental analysis as well as HRMS confirms the compound and purity. We have attached a new NMR for suggested compounds at high temperature with updated supplementary information. Also, we took NMR of compound no. 5o at 80 °C to confirm the rotamers and attached in the updated supporting information.

Comment: The numbering of the molecules in the supporting information (1-19) is different from that given in the paper (5a-5r).

Response: We appreciate your diligence in reviewing our supplementary section and we apologize for the discrepancy in the numbering. We have clarified this discrepancy by ensuring consistency in numbering between our main text and supporting information for clear understanding.

Comment The 1H NMR given in the supplementary material (and with the differences observed in the elemental analysis) don’t seem to be very pure. If there are issues with the NMR maybe HPLC should address the purity. 

Response: Thank you for your genuine response and kind suggestion. Our compounds are forming a hydrate with water since elemental analysis has a water molecule. NMR in the supplementary information are pure but because of the rotamers it the peaks are in some cases broad.

Closing Statement

We sincerely appreciate your constructive feedback which has greatly improved our manuscript's quality. Thank you once again for your time and consideration; we look forward to your further feedback on our revised submission. Feel free to adjust any specific wording or details as needed!

Round 2

Reviewer 1 Report

Comments and Suggestions for Authors

The quality of the manuscript has been significantly improved after revision, it should be accepted.

Author Response

Comment 1: The quality of the manuscript has been significantly improved after revision, it should be accepted.
Response: We are very grateful for the Reviewer's effort and commitment in assessing our manuscript.

Closing Statement

We sincerely appreciate your valuable feedback and suggestions, which have significantly improved our manuscript. Thank you for your consideration, and we look forward to your further feedback on our revised submission. Feel free to adjust any specific wording or details as needed!

Reviewer 2 Report

Comments and Suggestions for Authors

I was again invited to review the manuscript “Synthesis, Antimicrobial Activity, and Tyrosinase Inhibition by Multifunctional 3,4-Dihydroxy-Phenyl Peptidomimetics” based on a previous submission. Some aspects of my previous review were followed and addressed. However, there are still some issues:

-In line 37, where it says “According to recent study in 2024, …” it should say “According to a recent study in 2024, …”

-In line 99, “A common characteristic among many of these inhibitors is the presence of hydroxy groups, particularly dihydroxy groups, in their chemical structures.” the word “hydroxy groups” is repeated twice.

-In line 130, the yields reported (ranging from 48 to 68%) are moderate. Good yields are often considered above 70% and yields above 50% are "fair" or “moderate”.

-In line 152, where it says, “which shows only 15% inhibition activity at 100 μM (Figure 1).” should it say figure 3?

-In figures 4a and 4d, the residues which are responsible of the interactions should be labeled in the 3D figures for an easier understanding of the interactions.

-Please note that in compounds 5j (line 465), 5o (line 510) and 5p (line 519) the “N” should be in italics.

-Also, in compounds 5j (line 466) and 5p (line 520) where it says 1H NMR, it should be 1H NMR.

-The constants of C-F (JC-F) in the 13C NMR of compound 5r (line 541) should be in italics.

Kind regards

Author Response

Response to Reviewer Comments

Comment 1: -In line 37, where it says “According to recent study in 2024, …” it should say “According to a recent study in 2024, …”.
Response: Thank you for your valuable feedback. Mentioned sentence was corrected according to the Reviewer suggestion  

Comment 2: -In line 99, “A common characteristic among many of these inhibitors is the presence of hydroxy groups, particularly dihydroxy groups, in their chemical structures.” the word “hydroxy groups” is repeated twice.
Response: Thank you for your valuable feedback. Mentioned sentence was corrected according to the Reviewer suggestion  

Comment 3: -In line 130, the yields reported (ranging from 48 to 68%) are moderate. Good yields are often considered above 70% and yields above 50% are "fair" or “moderate”.
Response: Thank you for your valuable feedback. Mentioned sentence was corrected according to the Reviewer suggestion

Comment 4: -In line 152, where it says, “which shows only 15% inhibition activity at 100 μM (Figure 1).” should it say figure 3?
Response: Thank you for your valuable feedback. Figure numbering was corrected according to the Reviewer suggestion

Comment 5 -In figures 4a and 4d, the residues which are responsible of the interactions should be labeled in the 3D figures for an easier understanding of the interactions.
Response: We appreciate your clarification on this point. Mentioned figures were modified according to the Reviewer suggestion.

Comment 6: -Please note that in compounds 5j (line 465), 5o (line 510) and 5p (line 519) the “N” should be in italics.
Response: This was corrected according to the Reviewer suggestion.

Comment 7: -Also, in compounds 5j (line 466) and 5p (line 520) where it says 1H NMR, it should be 1H NMR.
Response: This was corrected according to the Reviewer suggestion.

Comment 8: -The constants of C-F (JC-F) in the 13C NMR of compound 5r (line 541) should be in italics.
Response: This was corrected according to the Reviewer suggestion.

Closing Statement

We sincerely appreciate your constructive feedback which has greatly improved our manuscript's quality. Thank you once again for your time and consideration; we look forward to your further feedback on our revised submission. Feel free to adjust any specific wording or details as needed!